# Understanding the COVID-19 Pandemic's Impact on E-Learner Satisfaction at the Tertiary Level

Mohammed Rafiqul Islam [1], Rimon Sarker [2], Rebaka Sultana [3], Md. Faisal-E-Alam [4],
Rui Alexandre Castanho [5,6,*] and Daniel Meyer [6]

1 Department of Management, Jatiya Kabi Kazi Nazrul Islam University, Trishal,
  Mymensingh 2220, Bangladesh
2 Department of Human Resource Management, Jatiya Kabi Kazi Nazrul Islam University, Trishal,
  Mymensingh 2220, Bangladesh
3 Department of Business Administration (Management), Bangladesh Army International University of Science
  and Technology (BAIUST), Cumilla Cantonment, Cumilla 3100, Bangladesh
4 Department of Management Studies, Begum Rokeya University, Rangpur 5404, Bangladesh
5 Faculty of Applied Sciences, WSB University, 41-300 Dabrowa Górnicza, Poland
6 College of Business and Economics, University of Johannesburg,
  P.O. Box 524, Auckland Park, Johannesburg 2006, South Africa
* Correspondence: acastanho@wsb.edu.pl

**Abstract:** E-learning has recently gained considerable interest among stakeholders, including educators, students, and policymakers. During the pandemic, organized online learning is critical to an effective e-learning system because it helps both teaching and learning. Thus, the current study intends to explore the factors contributing to e-learners' satisfaction during the COVID-19 pandemic. A questionnaire survey was conducted to gather data from 650 university students selected through convenience sampling. The data were analyzed using the Statistical Package for the Social Sciences (SPSS). The factors essential to boosting e-learner satisfaction were identified using confirmatory factor analysis (CFA). Frequency distribution and percentages were used to identify the demographic characteristics of respondents, and a reliability test was conducted to test the internal consistency of the data. This study employed structural equation modeling (SEM) to trace the relationship between the six independent variables and e-learner satisfaction. Regression results revealed that psychological factors, educational materials and design, access to technological devices, instructor attributes, and perceptions and expectations significantly influence e-learner satisfaction. However, students' engagement had no significant influence on the same. Because, most respondents had a clear preference for physical learning. The findings of this study will help educationists and policymakers take necessary steps in enhancing learners' satisfaction and improving their academic performance.

**Keywords:** e-learning; COVID-19; student's satisfaction; structural equation modeling (SEM)

## 1. Introduction

The coronavirus (COVID-19) that first appeared in the city of Wuhan in China in December 2019 has adversely affected livelihoods and the economy globally [1]. Worldwide educational institutions have also encountered an unprecedented crisis. The United Nations Education, Scientific, and Cultural Organization (UNESCO) acknowledged that the pandemic has confused almost 300 million learners about the continuation of their academic activities and has hampered their plans [2]. The entire system of education has entered a new phase called e-learning. To minimize the effect of the pandemic on the education sector, most developed countries' educational institutions have adopted e-learning [3].

COVID-19 significantly affected education across countries, such as the United States, including universities and other educational institutions. E-learning is a crucial part of

teaching methodology. According to studies, time-saving, the opportunity to attend a university far from home (in another city or country), the capacity to balance employment and study, and lower transportation costs are the benefits of remote learning that students find most valuable [4].

Other studies, however, have found that, while students were initially interested and satisfied with their in-person learning, those same qualities decreased during remote instruction. When learning remotely, undergraduate students felt more irritated, less accountable, and less engaged, and when the semester came to a close, they began to suffer academically [5]. However, a different study found that students' experiences at participating universities varied significantly. The "home learning environment", "engagement", and the "judgment of impact on learning skills" were the areas with the greatest variations. The "home learning environment" varied depending on the economic and technological progress of the three countries studied: South Africa, Wales, and Hungary [6].

Many developing and underdeveloped countries have failed to apply e-learning because of insufficient infrastructure, logistics support, and sophisticated technology [7]. The government of Bangladesh closed all educational institutions in March 2020 and suspended all offline academic activities during the COVID-19 shutdown. Lockdowns have made it difficult to change the academic calendar's activities. Teachers and students are more comfortable with physical classes, but the COVID-19 pandemic has forced them to adopt e-learning. Such a rapid change in online platforms has affected most students and teachers in several ways [8]. All these changes have challenged the quality of the online education system and learners' satisfaction—a core component of any healthy education system.

In Bangladesh, three categories of universities—public, private, and international—are operated under the University Grant Commission (UGC). Public universities run activities mainly with government funds. Students of public universities are from diverse economic, social, and geographical backgrounds. Because of this, it was not easy to implement e-learning at public universities. However, there are several prerequisites for the smooth functioning of the e-learning system. A deliberative and realistic policy, appropriate technology, and trained faculties are the prerequisites for the proper implementation of e-learning [9]. However, providing sophisticated technology is not sufficient. The easy and flexible use of technology by teachers and learners is more important in developing a healthy e-learning environment.

On the other hand, private universities depend on self-generated funds mainly collected as tuition fees from students. Students of this university category come from middle-class or wealthy families and they are capable of managing devices necessary for e-learning. Thus, it is necessary to assess the state of e-learning in Bangladesh's private universities and to know the factors contributing to learners' satisfaction in the country. Therefore, the main objective of this study is to determine the factors contributing to e-learner satisfaction at private universities during the COVID-19 pandemic.

E-learning is mostly a new concept in Bangladesh's education sector. As a result, additional research is being performed to fully comprehend the benefits, drawbacks, and difficulties of this kind of educational institution [10]. The uniqueness of the current research method lies in the identification of the dimensions (psychological factors, educational materials and design, access to technological devices, learner engagement, instructor attributes, and perceptions and expectations) of student evaluations of remote learning in higher education.

While higher education administration, teachers, and students tried to figure out how to organize, teach, study, and sustain a campus culture through a remote learning platform, this decision was taken without precedent or research outlining the best practices to follow. Despite being a growing discussion issue, very little is known regarding the effects of COVID-19 on higher education [11,12]. The dearth of studies on this subject confirms the significance of studies such as this one.

Moreover, the United Nations' Sustainable Development Goals (SDGs) are a set of 17 goals aimed at achieving a sustainable future for all. SDG4 focuses on ensuring inclusive and equitable quality education and promoting lifelong learning opportunities for all. E-learning offers several benefits, including flexibility, accessibility, and cost-effectiveness for all regardless of their socioeconomic background, gender, or geographical location. E-learning provides an opportunity to bridge the education gap and provide access to quality education for everyone. Given the current significant transition to digital education caused by the COVID-19 pandemic, e-learning can be seen as a potential way to achieve sustainability in education [13].

## 2. Literature Review and Hypothesis Development

COVID-19 has affected the education system worldwide and was the reason schools and universities closed [14]. Therefore, e-learning has become an important issue in the education process [15]. During the COVID-19 pandemic, universities throughout the world witnessed a significant evolutionary shift toward the e-learning system. As online-based education systems helped teaching–learning during the COVID-19 pandemic on an ad hoc basis, implementing a properly planned and structured online-based education process is essential for an effective e-learning system [16]. This section summarizes the evidence from existing literature concerning six factors considered in this study, followed by connected hypotheses related to those factors.

### 2.1. Psychological Factors

Deaths of nears and dears, movement restrictions, confinement at home, financial hardship, and other adverse effects due to the COVID-19 pandemic have undoubtedly affected learners' psychology. A good mental state in learners is a prerequisite for effective e-learning. Family, friends, and teachers provide a supportive environment and encourage students in their e-learning classes [17]. Kemp and Grieve [18] studied two separate groups of students and found them to be interested in offline classroom discussions, but they preferred online assignments and projects. However, the pandemic's impact on student life satisfaction and depression, anxiety, and stress indices was particularly notable [19]. Collaboration and social interactions with teachers and friends made students psychologically better able to go through the e-learning system. Shih et al. [20] stated that positive groups improved communication and cooperation between friends and provided mental support to learners. Thus, the hypothesis concerning psychological factors is as follows:

**H1.** *Psychological factors influence e-learner satisfaction.*

### 2.2. Educational Materials and Design

A well-designed course, supporting materials, and sophisticated technologies facilitate students' e-learning [21]. In this way, the authors [22] have suggested that course materials should be tailored to students' capabilities. In addition, online-based pedagogy and course designs for higher education must be student-centered rather than teacher-centered [23]. Sometimes, the online learning system is better than the physical one because of time-saving and quick arrangements [24]. Furthermore, educational materials and course designs supported by sophisticated technologies, multidisciplinary resources, and flexible and understandable teaching–learning concepts make the environment more interesting for e-learners [25]. Students' replies revealed that they were pleased with instructional video information and that they had learned from them [26]. Updated design for e-learning course materials supports an exciting and challenging environment through the teaching–learning process [27]. Thus, the hypothesis concerning educational materials and design is as follows:

**H2.** *Educational materials and design influence e-learner satisfaction.*

### 2.3. Access to Technological Devices

Sophisticated technological connections are inevitable in online courses offered by universities, colleges, and schools [28,29]. E-learning conformance apps enable students to turn to online courses for more fun and satisfaction [30]. Additionally, students and teachers need to be trained in appropriate technical skills before moving to the online-based education system [31,32]. Demuyakor [33] found internet connectivity to be a significant barrier to continuing education in online platforms. According to the findings, technological devices are the key factor in boosting students' learning satisfaction and performance [34]. Further, the amount of student satisfaction with their learning in the e-learning environment is significantly influenced by technological aspects [35]. The findings of one study demonstrated how user satisfaction and e-learning preparation were strongly influenced by technological competency [36]. Access to internet facilities also has a significant impact on learners' satisfaction. Thus, the hypothesis concerning access to technological devices is as follows:

**H3.** *Access to technological devices influences e-learner satisfaction.*

### 2.4. Learner Engagement

Learner engagement ensures the quality and quantity of a learner's participation in every aspect of education system. Educationists have explored the ways students become satisfied with online courses if they have freedom of choice [37]. According to [38,39], the determinants of a successful e-learning process include an engaging teaching–learning environment, better student communication skills, understanding communication between teachers and students, appropriate course designs, quality course contents, and administrative support service. In a physical learning approach, the interaction between teachers and students happens face-to-face [40], but the e-learning process is not sufficiently prepared for students to participate [41]. However, learner engagement enriches communication skills and ensures quality, which increases overall satisfaction [42]. Thus, the hypothesis concerning learner engagement is as follows:

**H4.** *Learner engagement influences e-learner satisfaction.*

### 2.5. Instructor Attributes

The role of teachers, along with an appropriately designed curriculum, makes the e-learning process effective [43]. Instructors should provide additional effort in creating an exciting environment by developing an online-based course outline rather than a physical teaching process [44]. According to [45], instructors' timely feedback is more important to the students for knowing their understanding level. As instructors are essential parts of education, observing their performance through peer evaluation ensures their efficiency and quality in the e-learning process [46]. A positive relationship between instructors and students makes a teaching–learning environment more satisfactory [47,48]. Giving students the chance to study in accordance with their unique learning styles, timetables, and locations requires the instructor's support in order for students to obtain a more thorough understanding of how to control their own learning processes [49]. Researchers such as [50–53] have found that the quality of a teacher's delivery, assessment techniques, and sophisticated technologies are significant determinants of e-learner satisfaction. Thus, the hypothesis concerning instructor attributes is as follows:

**H5.** *Instructor attributes influence e-learner satisfaction.*

### 2.6. Perceptions and Expectations

Students' perceptions and expectations are essential determinants for ensuring e-learner satisfaction. Reeve and Jang [54] noted five elements of student satisfaction: relevance to learning, training, authentic learning, self-directed learning, and technical experience. Kuo et al. [55] found that the interaction between students and teachers using

technology affected students' positive perceptions regarding e-learning. Binyamin, Rutter and Smith [56] observed that students' perceptions and expectations are essential to effectively influence the satisfactory level of online education. The authors concluded that satisfaction with e-learning varies due to undergraduates, postgraduates, low-experience students, and high-experience students. Yu et al. [57] found interactions between students and teachers are closely connected to learning promises in the online learning environments. Users' expectations of how enjoyable, practical, and simple to use e-learning apps must be realized if they are to be effective [58]. Perceptions and expectations influence user satisfaction in e-learning [59]. Thus, the hypothesis concerning perceptions and expectations is as follows:

**H6.** *Perceptions and expectations influence e-learner satisfaction.*

Learners' satisfaction with the learning process is the engine of success for any education system. All previous studies were based on a single variable, few variables, or overall e-learning systems, whereas the current research considered several items together. In this regard, the unique input of this research is the extraction of two factors, that is, psychological factors and perceptions and expectations. In addition, most of the previous studies have used regression models, but this study was conducted using structural equation modeling (SEM). Further, the proposed framework in the current research will mitigate the contextual gap in the educational sector, as shown in Figure 1.

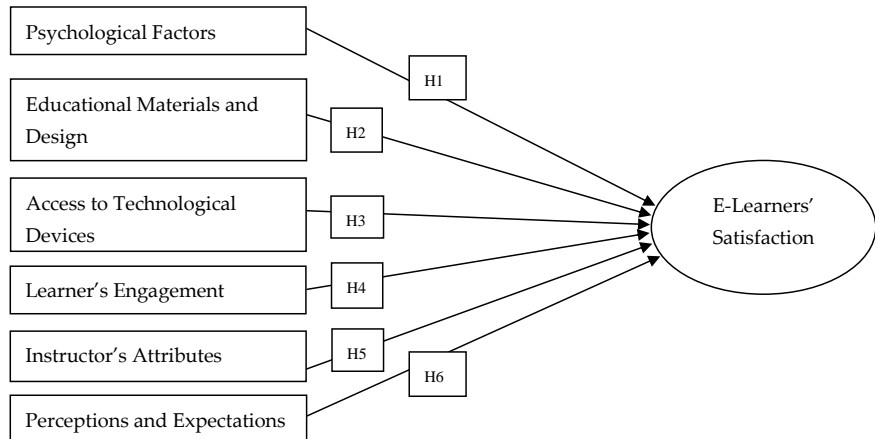

**Figure 1.** Conceptual framework.

## 3. Materials and Methods

The current descriptive study is based on quantitative data collected from June 2020 to July 2021. Descriptive research involves analyzing and interpreting the characteristics of a situation or condition, or determining whether relationships exist between variables [60]. A survey is employed for descriptive research design [61]. This study used the survey method to gather primary data from the respondents by making a structured questionnaire.

### 3.1. Research Instruments

The questionnaire used in this study comprised 34 informational items. Of these, 9 were related to demographic information, such as gender, age, education, worries about study gaps, willingness to use online courses to avoid study delay, types of electronic devices, the nature of internet connectivity, daily time spent on the internet, and time spent for education on the internet. The remaining 25 items were related to e-learner satisfaction, including fear of the virus, lack of communication, depression, course content, course design, course flexibility, device availability, internet quality, accessibility, learner attitudes, learner expertise, learner timeliness, learner concentration, instructor attitude, instructor timeliness, instructor patience, instructor mentoring, good ideas, usefulness, improvements, home environments, parent pressure, peer pressure, evaluation, and satisfaction. The

questionnaire uses a 5-point Likert scale, from 1 (strongly disagree) to 5 (strongly agree), to measure the positive or negative perceptions of the statements.

### 3.2. Administering the Questionnaire

For measuring the impact of COVID-19 on e-learner satisfaction, data were collected through an online questionnaire survey on 650 students studying at Bangladesh's six private universities. Universities, as well as respondents, were selected following the convenience sampling approach. The respondents were undergraduate and postgraduate students continuing their education through online classes. The current study considered six independent factors used in previous studies: psychological factors [20], educational materials and design [22], access to technological devices [33], learner engagement [38], instructor attributes [46], and perceptions and expectations [57]. These variables were tested to examine their relationship with e-learner satisfaction. A pilot survey was also conducted on 30 respondents to test the questionnaire's reliability and avoid any ambiguity.

### 3.3. Data Analysis Techniques

The Statistical Package for the Social Sciences (SPSS) was employed to analyze the data. Confirmatory factor analysis (CFA) was used to determine the factors playing vital roles in enhancing e-learner satisfaction. Frequency distribution and percentages were used to identify the demographic characteristics of the respondents. A reliability test was conducted to test the internal consistency of the data. Furthermore, SEM was used to find the relationship between six independent variables and e-learner satisfaction. In addition, a multiple regression model was also applied to determine the relationship between the independent variables and e-learner satisfaction (dependent variable).

## 4. Results

### 4.1. Demographic Statistics

As shown in Table 1, the number of female respondents was slightly higher than male respondents, and the age of more than 85% of respondents was between 20 and 30 years. Regarding education, 61.54% of respondents were undergraduate students, and the remainders were postgraduate students in sample private universities. Information concerning respondents' worries about the study gap, interest in taking online courses, types of electronic devices used, kind of internet connectivity, time spent on the internet per day, and time spent on the internet for academic purposes are also summarized in Table 1.

**Table 1.** Demographic characteristics of respondents (*n* = 650).

| Demographic Factors | Freq. | Pert |
| --- | --- | --- |
| Gender | | |
| Male | 300 | 46.15 |
| Female | 350 | 53.85 |
| Age | | |
| Below 20 | 18 | 2.77 |
| 20–25 | 336 | 51.69 |
| 26–30 | 220 | 33.85 |
| Above 30 | 76 | 11.69 |
| Education | | |
| Graduate level | 400 | 61.54 |
| Postgraduate level | 250 | 38.46 |
| Worried about the study gap caused by COVID-19 | | |
| Yes | 110 | 16.92 |
| No | 460 | 70.77 |
| Undecided | 80 | 12.31 |

**Table 1.** *Cont.*

| Demographic Factors | Freq. | Pert |
|---|---|---|
| Want to take online courses to avoid study delay | | |
| Yes | 390 | 60 |
| No | 210 | 32.31 |
| Undecided | 50 | 7.69 |
| Types of accessible electronic devices | | |
| Computer without Webcam | 180 | 27.69 |
| Computer with webcam | 10 | 1.54 |
| Smartphone only | 380 | 58.46 |
| Tablet/IPad only | 20 | 3.07 |
| Both computer and smartphone | 60 | 9.23 |
| None of the above | 0 | 0 |
| Kinds of internet connectivity used | | |
| Broadband | 30 | 4.62 |
| Mobile data | 595 | 91.54 |
| Both broadband and mobile data | 25 | 3.85 |
| None of the above | 0 | 0 |
| Time spent on the internet per day | | |
| Less than 1 h | 0 | 0 |
| 1–3 h | 100 | 15.38 |
| 4–5 h | 400 | 61.54 |
| 6–7 h | 150 | 23.07 |
| Time spent on education on the internet | | |
| Less than 1 h | 0 | 0 |
| 1–3 h | 150 | 23.07 |
| 4–5 h | 350 | 53.85 |
| 6–7 h | 150 | 23.07 |

### 4.2. Validity and Reliability of Measurement Model

There are two ways to analyze convergent validity. In the first method, the factor loadings evaluate the validity of the constructs and the second method is applied to analyze composite reliabilities [62]. Moreover, reliability analysis helps determine the seven determinants of Cronbach's alpha, which ranged from 0.877 to 0.981. A Cronbach's alpha value greater than 0.6 for all items indicates an acceptable level for the factor loadings (FL) [63]. The findings show convergent validity and reliability test performed in AMOS Software, which helped us comprehend the determinants under each construct (see Table 2).

**Table 2.** Results of measurement model—convergent validity.

| Construct | Items | Factor Loadings (FL) | Construct Reliability | Kaiser–Meyer–Olkin (KMO) | Average Variance Extracted (AVE) |
|---|---|---|---|---|---|
| Psychological Factors | PF1 | 0.947 | 0.952 | 0.716 | 0.914 |
| | PF2 | 0.972 | | | |
| | PF3 | 0.936 | | | |
| Educational Materials and Design | EMD1 | 0.875 | 0.883 | 0.690 | 0.827 |
| | EMD2 | 0.900 | | | |
| | EMD3 | 0.946 | | | |
| Access to Technological Devices | ATD1 | 0.975 | 0.981 | 0.737 | 0.964 |
| | ATD2 | 0.970 | | | |
| | ATD3 | 0.987 | | | |

**Table 2.** *Cont.*

| Construct | Items | Factor Loadings (FL) | Construct Reliability | Kaiser–Meyer–Olkin (KMO) | Average Variance Extracted (AVE) |
|---|---|---|---|---|---|
| Learner Engagement | LE1 | 0.942 | 0.877 | 0.641 | 0.737 |
| | LE2 | 0.671 | | | |
| | LE3 | 0.846 | | | |
| | LE4 | 0.901 | | | |
| Instructor Attributes | IA1 | 0.977 | 0.903 | 0.697 | 0.968 |
| | IA2 | 0.972 | | | |
| | IA3 | 0.899 | | | |
| | IA4 | 0.944 | | | |
| Perceptions and Expectations | PE1 | 0.958 | 0.927 | 0.774 | 0.820 |
| | PE2 | 0.952 | | | |
| | PE3 | 0.960 | | | |
| | PE4 | 0.658 | | | |
| E-Learner Satisfaction | ES1 | 0.918 | 0.935 | 0.667 | 0.837 |
| | ES2 | 0.914 | | | |
| | ES3 | 0.911 | | | |
| | ES4 | 0.904 | | | |

In addition, when the average variance extracted (AVE) was greater than 0.5 for each construct, all the determinants were acceptable for the analysis [64]. Furthermore, when the Kaiser–Meyer–Olkin (KMO) test result is greater than 0.5, that means it is acceptable, and in this research work, the KMO value ranged from 0.641 to 0.774, which is highly useful for analysis (see Table 2).

*4.3. Structural Equation Modelling (SEM)*

SEM helps evaluate the actual relationship between the exploratory factor analysis (EFA) and multiple regression [65]. In this study, confirmatory factor analysis (CFA) is applied as a replacement for EFA in measuring the data fitness of hypothesized measurement model [66]. In addition, it is acceptable when the eigenvalue is greater than one in the factor analysis. Furthermore, varimax rotation was employed to measure the approachable determinants. Factor analysis is more effective in research that intends to determine the relationship between the determinants. Only variables with loadings greater than 0.5 are considered acceptable in factor analysis, and the lower loading values are removed. Based on the factor analysis, the constructs in the model are shown in Figure 2.

*4.4. Coefficient of Determination*

Similar to [67], the current study applied the coefficient of determination ($R^2$) test. This model helps to show accurate data predictions, which provide considerable contributions [68]. As all the construct validity results showed $R^2$ values greater than 0.70, they were considered acceptable for analysis. The hypothesis test results are shown in Table 3, based on the analysis.

**Table 3.** Summary result of testing hypotheses.

| Hypotheses | Beta | *t*-Value | *p*-Value | Decision |
|---|---|---|---|---|
| **H1**. Psychological factors influence e-learner satisfaction | 3.530 | 9.916 | 0.003 | Supported |
| **H2**. Educational materials and design influence e-learner satisfaction | 3.424 | 23.371 | 0.000 | Supported |
| **H3**. Access to technological devices influences e-learner satisfaction | 3.442 | 23.659 | 0.000 | Supported |
| **H4**. Learner engagement influences e-learner satisfaction | 3.003 | 22.487 | 0.352 | Not Supported |
| **H5**. Instructor attributes influence e-learner satisfaction | 3.750 | 24.503 | 0.000 | Supported |
| **H6**. Perceptions and expectations influence e-learner satisfaction | 3.442 | 21.603 | 0.000 | Supported |

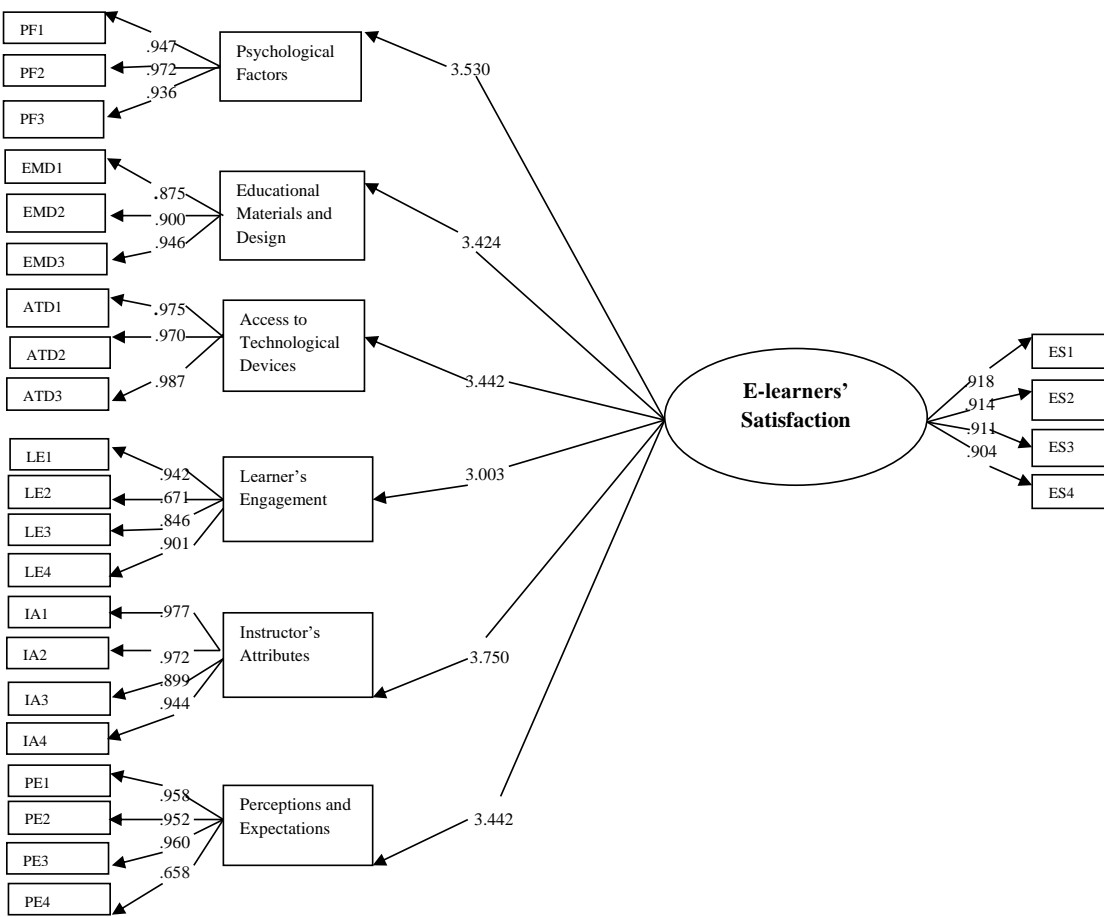

**Figure 2.** Structural equation modeling—CFA.

## 5. Discussion

This study has identified instructor attributes as one of the most influential determinants of e-learner satisfaction. The instructor's teaching method, course materials, flexibility, time management, and practical knowledge are a few attributes that significantly affect the design of individual courses. The commitment and seriousness of instructors also affect the mindset of students. The instructor aspect can be seen to have a positive and significant relationship with e-learner satisfaction [69]. The study also found that psychological factors and educational materials and design significantly impact e-learner satisfaction. However, one study result did not agree with this outcome, in that psychological factors failed to influence e-learning satisfaction [70]. However, the findings show a statistically significant correlation between e-learning materials and quality and student satisfaction [71,72].

The study revealed that e-learner satisfaction largely depends on the appropriate application of the system. Without adequate planning and course design, there will be more difficulties in online classes, ultimately hampering the education system. Thus, the role of an instructor is essential for the successful implementation of e-learning systems. The study also revealed that access to technological devices and perceptions and expectations have an appreciable influence on e-learner satisfaction. Thus, learners with limited virtual competency may experience greater challenges and obtain lower satisfaction than others [73]. Likewise, students' perceptions and expectations of online learning are largely positive [74,75]. In addition, device availability and internet connectivity for teachers and students are the prime prerequisites for an effective e-learning system. According to the findings, perceived utility, perceived simplicity of use, system and technical aspects, and instructor traits are the most important elements influencing learners' perceived satisfaction [76].

However, this research found an insignificant and negative connection between learner engagement and e-learner satisfaction. Similarly, an investigation of connection between virtual learning engagement and satisfaction found insignificant in another study [77]. However, another study's findings disagree, in that engagement and satisfaction were significantly connected and had a positive relationship [78]. One of the most important reasons is that students have different technological devices not used for academic purposes; instead, they are addicted to online games and social media activities. For this reason, most students are less engaged in their academic activities, which may negatively affect e-learning. Therefore, teachers' real-world knowledge sharing with students makes the teaching–learning environment interesting and thus helps students engage more in e-learning. The results also contradict a little bit with other studies where willingness was found to influence the learner's satisfaction, along with the technological aspect and the design aspect [79].

E-learning setups can resolve difficult and real-world complications, make the learning process more efficient, obtain prognostic exhibitions and real-time conceptions, and help students retain more knowledge than the traditional method. Technology-based learning are key to stimulating students, addressing knowledge gaps, reducing costs of teaching, collaborating with financial aspects of education, and sharing resources and infrastructure with future leaders [80]. The study findings are related to the study outcomes, which revealed that learner computer anxiety, instructor attitudes, e-learning course flexibility, e-learning course quality, perceived usefulness, perceived ease of use, and diversity in evaluations are the main aspects of e-learning satisfaction [81].

This study contributes to the existing literature because of its emphasis on the most demanding issues in the COVID-19 pandemic situation, providing a solution by focusing on significant determinants such as psychological factors, educational materials and design, access to technological devices, instructor attributes, and perceptions and expectations. Further, it may improve different technology-based platforms, creating a vast opportunity for students to ensure effective e-learning. These findings show that developing countries may be able to meet Sustainable Development Goal 4 (Education) by enhancing learner satisfaction and boosting the application of their e-learning strategies [69]. For this reason, e-learner satisfaction is a key factor in achieving SDG4. If learners are satisfied with the e-learning experience, they are more likely to continue their education and achieve their goals.

## 6. Conclusions

The COVID-19 pandemic has created enormous challenges for universities and made them unable to continue the standard procedures. E-learning emerged as an essential part of an effective education system during the COVID-19 pandemic. The present study has provided empirical evidence of critical student problems by highlighting six factors contributing to e-learner satisfaction. Instructor attributes were identified as the most significant factor positively influencing e-learner satisfaction. In addition, a lack of student engagement was identified as the main concern in an effective e-learning system. The study found that a significant number of students could not concentrate on online classes because of insufficient resources. Although student feedback is a vital component in ensuring an effective teaching–learning process, this study found it to be a major hurdle for teachers.

This study will provide valuable input to universities, policymakers, and governments for designing/redesigning effective e-learning systems in any pandemic situation. The research findings will help academicians improve the effectiveness of the e-learning system and enrich the quality of universities' teaching–learning systems. Universities should encourage students to leave online reviews and solicit their comments in moving toward e-learning systems. The government should also offer affordable internet packages and smartphone programs. Though the study was designed scientifically to ascertain insights concerning e-learning, it has some limitations. First, the study examined the factors contributing to e-learner satisfaction based on a survey of a few private universities in



Bangladesh. The results could be more comprehensive and representative if the number of institutions was increased. Second, this study is based on the survey of student opinions only. Thus, future researchers can focus on students' and teachers' perceptions to gain more profound knowledge about e-learner satisfaction.

**Author Contributions:** Conceptualization, M.R.I. and R.S. (Rebaka Sultana); methodology, R.S. (Rebaka Sultana), M.R.I. and R.S. (Rimon Sarker); software, M.R.I. and R.S. (Rebaka Sultana); validation, M.F.-E.-A., R.A.C. and D.M.; formal analysis, R.S. (Rebaka Sultana) and M.R.I.; investigation, M.R.I. and R.S. (Rebaka Sultana); resources, R.S. (Rimon Sarker), D.M. and R.A.C.; data curation, D.M., M.F.-E.-A. and R.A.C.; writing—original draft preparation, M.R.I., R.S. (Rebaka Sultana) and R.S. (Rimon Sarker); writing—review and editing, M.R.I., R.S. (Rimon Sarker) and M.F.-E.-A.; visualization, R.A.C., M.F.-E.-A. and D.M.; supervision, R.A.C. All authors have read and agreed to the published version of the manuscript.

**Funding:** This research received no external funding.

**Institutional Review Board Statement:** Not applicable.

**Informed Consent Statement:** Not applicable.

**Data Availability Statement:** Not applicable.

**Conflicts of Interest:** The authors declare no conflict of interest.

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
