# Peer review of "Understanding the COVID-19 Pandemic’s Impact on E-Learner Satisfaction at the Tertiary Level"

_sustainability, doi:10.3390/su15086694_

Round 1
Reviewer 1 Report
Please see attachment

Author Response
Response Sheet-Reviewer-01

Reviewer 2 Report
Change the expression "during the pandemic..." and "of the pandemic..." on pages 2, 3, 10, 11, to "COVID-19 pandemic" for consistency in reading.
All table descriptions are at the top, not at the bottom. Check the width of the tables that exceed the established width.
Also the acronyms, abbreviations or acronyms used, for example missing to specify factor loadings (FL) of table 2 in the previous paragraph.
Figure 1 and Table 3 are not mentioned in the preceding or following paragraphs.
In Figure 2, the weights can be outside the connector lines to distinguish the point from the factor loadings (first level) and from the third level which do not have an integer value (zero can be assumed to identify the integer) and from the second level which has an integer value. Also, as it is specified "Only the variables with loadings greater than 0.5 are considered acceptable", where it is written with the integer zero, zeros can be placed.
Finally, suspensive points must be placed in figure 2 in order not to think that it is a sum of weights to the trajectory.
Author Response
Response Sheet Reviewer-02

Reviewer 3 Report
Thank you for a very interesting and timely study with many future implications. I found it very informative and interesting to read. My comments go to improving the paper's contribution and fall into related substantive areas.
Introduction
· There is some repetition in the context; the discussion can be provided more tightly, highlighting why research is needed in this particular area.
· The red text from lines 93-101 on higher education and sustainability came completely out of the blue and did not fit here. A separate section before the literature review would help provide a relevant narrative and link between the key terms. Some discussion on SDG4 (quality education) would be useful here.
Discussion
My suggestion is to develop a link with SDG4; this would align the paper with the journal's scope more closely. I feel a Reflective section (i.e., Reflection) after discussion would be extremely useful for the reader. This section must reflect on the findings and quality of education (SDG4), especially how this paper can contribute to the target goals of SDG4.
Minor issues
Several minor issues were noted. Do a full and detailed ref, spelling, and typing check of the paper.
Overall comments
A more robust approach must be adopted to discuss sustainability and higher education, especially concerning the context of this study.
This is a promising study. I hope my comments will prove helpful to you. Good luck!
Author Response
Response Sheet Reviewer-03

Round 2
Reviewer 1 Report
The authors have complied with all comments and significantly improved the quality of the manuscript. I recommend acceptance of the manuscript for publication.